# Recent evolutionary origin and localized diversity hotspots of mammalian coronaviruses

Renan Maestri[1,2]*[†], Benoît Perez-Lamarque[1,3][†], Anna Zhukova[4], Hélène Morlon[1]

[1]Institut de Biologie de l'École Normale Supérieure (IBENS), École Normale Supérieure, CNRS, INSERM, Université PSL, Paris, France; [2]Departamento de Ecologia, Instituto de Biociências, Universidade Federal do Rio Grande do Sul, Porto Alegre, Brazil; [3]Institut de Systématique, Évolution, Biodiversité (ISYEB), Muséum national d'histoire naturelle, CNRS, Sorbonne Université, EPHE, UA, Paris, France; [4]Institut Pasteur, Université Paris Cité, Bioinformatics and Biostatistics Hub, Paris, France

*For correspondence: renanmaestri@gmail.com

[†]These authors contributed equally to this work

Competing interest: The authors declare that no competing interests exist.

**Abstract** Several coronaviruses infect humans, with three, including the SARS-CoV2, causing diseases. While coronaviruses are especially prone to induce pandemics, we know little about their evolutionary history, host-to-host transmissions, and biogeography. One of the difficulties lies in dating the origination of the family, a particularly challenging task for RNA viruses in general. Previous cophylogenetic tests of virus-host associations, including in the Coronaviridae family, have suggested a virus-host codiversification history stretching many millions of years. Here, we establish a framework for robustly testing scenarios of ancient origination and codiversification *versus* recent origination and diversification by host switches. Applied to coronaviruses and their mammalian hosts, our results support a scenario of recent origination of coronaviruses in bats and diversification by host switches, with preferential host switches within mammalian orders. Hotspots of coronavirus diversity, concentrated in East Asia and Europe, are consistent with this scenario of relatively recent origination and localized host switches. Spillovers from bats to other species are rare, but have the highest probability to be towards humans than to any other mammal species, implicating humans as the evolutionary intermediate host. The high host-switching rates within orders, as well as between humans, domesticated mammals, and non-flying wild mammals, indicates the potential for rapid additional spreading of coronaviruses across the world. Our results suggest that the evolutionary history of extant mammalian coronaviruses is recent, and that cases of long-term virus–host codiversification have been largely over-estimated.

## eLife assessment

Maestri et al report the absence of phylogenetic evidence supporting codiversification of mammalian coronaviruses and their hosts, leading to the **important** conclusion that the evolutionary history of the virus and its hosts are decoupled through frequent host switches. The evidence for frequent host switching, derived from state-of-the-art probabilistic modeling of co-evolution, is **convincing**. The study adds a new perspective to the ongoing debate over the timescale of coronavirus evolution.

## Introduction

Coronaviruses are RNA-viruses of the family Coronaviridae, comprising positive-sense and single-stranded viruses that have the largest genomes among nidoviruses (*Alekseev et al., 2008*; *De Groot, 2022*). As with several other RNA viruses, they may cause diseases in humans and other animals (*King*

**eLife digest** The SARS-CoV-2 virus, which caused the recent global coronavirus pandemic, is the latest in a string of coronaviruses that have caused serious outbreaks. This group of coronaviruses can also infect other mammals and likely jumped between species – including from non-humans to humans – over the course of evolution.

Determining when and how viruses evolved to infect humans can help scientists predict and prevent outbreaks. However, tracking the evolutionary trajectory of coronaviruses is challenging, and there are conflicting views on how often coronaviruses crossed between species and when these transitions likely occurred. Some studies suggest that coronaviruses originated early on in evolution and evolved together with their mammalian hosts, only occasionally jumping to and from different species. While others suggest they appeared more recently, and rapidly diversified by regularly transferring between species.

To determine which is the most likely scenario, Maestri, Perez-Lamarque et al. developed a computational approach using already available data on the genetics and evolutionary history of mammals and coronaviruses. This revealed that coronaviruses originated recently in bats from East Asia and Europe, and primarily evolved by rapidly transferring between different mammalian species. This has led to geographical hotspots of diverse coronaviruses in East Asia and Europe.

Maestri, Perez-Lamarque et al. found that it was rare for coronaviruses to spill over from bats to other types of mammals. Most of these spillovers resulted from coronaviruses jumping from bats to humans or domesticated animals. Humans appeared to be the main intermediary host that coronaviruses temporarily infected as they transferred from bats to other mammals.

These findings – that coronaviruses emerged recently in evolution, jumped relatively frequently between species, and are geographically restricted – suggest that future transmissions are likely. Gathering more coronavirus samples from across the world and using even more powerful analysis tools could help scientists understand more about how these viruses recently evolved. These insights may lead to strategies for preventing new coronaviruses from emerging and spreading among humans.

*et al., 2012*). Depending on the taxonomic arrangement, seven (*Leao et al., 2022*; *Wardeh et al., 2021*; *Zmasek et al., 2022*) or eight (*Becker et al., 2022*) species of coronaviruses infect humans, three of which being pathogenic: the SARS-CoV (*Drosten et al., 2003*; *Peiris et al., 2003*), the MERS-CoV (*Zaki et al., 2012*), and the SARS-CoV-2 (*Zhou et al., 2020*). The latter is at the origin of the recent COVID-19 pandemic that infected more than 775 million people and caused the death of more than seven million (*Ritchie, 2024*). Coronaviruses' high frequency of recombination (*Woo et al., 2010*), broad host range, and high mutation rates (*Becker et al., 2022*) make them especially prone to causing yet future diseases. Nevertheless, their evolutionary history and biogeography are very poorly understood. Resolving the evolutionary origins of Coronaviridae, understanding how they diversified, and characterizing their geographic diversity patterns would facilitate attempts to predict future zoonoses (*Becker et al., 2022*; *Anthony et al., 2017*; *Munoz, 2022*).

Coronaviruses infect mammals, birds, and fishes (*De Groot, 2022*), although they predominate in mammalian species (*Guan et al., 2003*; *Chinese SARS Molecular Epidemiology Consortium, 2004*; *Graham and Baric, 2010*; *de Groot et al., 2013*; *Corman et al., 2018*). A consensus exists on the taxonomic segregation of four genera within Coronaviridae: Orthocoronavirinae, namely Alpha-, Beta-, Gamma- and Deltacoronavirus (*De Groot, 2022*; *Mavrodiev et al., 2020*). Alpha- and betacoronaviruses are found exclusively in mammals, while delta- and gammacoronaviruses infect mostly birds but also mammals to a lesser extent (*Corman et al., 2018*; *Woo et al., 2009*; *Woo et al., 2012*). Coronaviruses are most numerous and genetically diversified in mammals (*De Groot, 2022*; *Woo et al., 2012*), in particular bats, suggesting a mammalian origin in bats (*De Groot, 2022*; *Corman et al., 2018*; *Woo et al., 2012*; *Vijaykrishna et al., 2007*), although this remains to be tested.

The timing of origination of the Coronaviridae family is debated, with results that vary by several orders of magnitude. *Woo et al., 2012* found a recent origin, around 10 thousand years ago. This dating was obtained by sequencing the well-conserved RNA-dependent RNA-polymerase (RdRp) genome region of representatives of all four coronavirus genera, and fitting to these sequences a neutral nucleotide-based substitution model with an uncorrelated log-normal relaxed clock

(*Drummond et al., 2006*) calibrated with serial samples. This calibration provided a mean substitution rate estimate of 1.3x10–4 substitutions per site per year. *Wertheim et al., 2013* used this estimate and the same genome region (RdRp), but with a codon-based substitution model accounting for the effect of selection. Indeed, purifying selection can lead to an underestimation of viral origins when not accounted for (*Wertheim et al., 2013*; *Wertheim and Kosakovsky Pond, 2011*). They found an ancient origin, around 293 (95% confidence interval, 190–489) million years ago (*Wertheim et al., 2013*). More recently, *Hayman and Knox, 2021* obtained similar results, but using the splitting times of hosts as constraints, therefore assuming a priori that coronaviruses codiversified with their hosts.

More generally, dating the phylogenies of RNA virus families is a difficult task (*Duchêne et al., 2014*). While for some of them dated calibration points can be used, based on orthologous copies of endogenous virus elements (EVEs) present in the genomes of related mammalian species with known times of divergence (*Katzourakis and Gifford, 2010*), in many others, including in the Coronaviridae, such elements have not been found (*Shi et al., 2018*). Despite the difficulty in dating viral families, it has been proposed, from cophylogenetic analyses investigating the congruence of the host and viral phylogenetic trees (*Conow et al., 2010*), that vertebrate-associated RNA viruses have codiversified with their hosts over hundreds of millions of years (*Shi et al., 2018*; *Zhang et al., 2018*; but see *Geoghegan et al., 2017*; see *Supplementary file 1a* for a clarification of out terminology). Indeed, RNA virus phylogenies tend to mirror that of their hosts; for example, closely-related coronaviruses infect closely-related mammals (e.g. *Hayman and Knox, 2021*). However, a major caveat is that such cophylogenetic signals can emerge when viruses diversify by host switches preferentially occurring among closely-related hosts, in the absence of any cospeciation event (*de Vienne et al., 2013*; *Perez-Lamarque and Morlon, 2024*).

Event-based cophylogenetic methods can in principle identify cospeciation and host switches events (*Conow et al., 2010*; *de Vienne et al., 2013*), but their behavior in the presence of diversification by preferential host switches is not well understood. Under a perfect codiversification scenario, host and symbiont phylogenies would be identical. Events of host switches, duplications and losses induce mismatches, and cophylogenetic methods aim to identify parsimonious sets of events that allow 'reconciling' the two phylogenies (*de Vienne et al., 2013*; *Perez-Lamarque and Morlon, 2022*). However, most of these methods rely entirely on tree topology (and not branching times), such that time-inconsistent host switches between non-contemporary host lineages are allowed during the reconciliation. In the presence of preferential host switches, these methods may thus favor biologically unrealistic reconciliations that involve cospeciation events and 'back-in-time' host switches to reconciliations that involve more frequent contemporary host switches. This would have remained unnoticed, unless users of the methods specifically looked at the time consistency of the inferred host switches, which is usually not done.

Here, we establish a framework for testing scenarios of ancient origination and codiversification versus recent origination and diversification by host switches that combines probabilistic cophylogenetic models and biogeographic analyses (*Figure 1*). We then apply this framework to the Coronaviridae-mammals association. We assemble a dataset of all mammalian hosts of coronaviruses and a complete association matrix between host species and species-like Operational Taxonomic Units (sOTUs) of coronaviruses, as well as geographic repartition of Coronaviridae and their mammalian hosts. We construct a new Coronaviridae tree based on a recent proposition for the use of a well-conserved region of their RNA genome (*Edgar et al., 2022*; *Babaian and Edgar, 2022*). Under the ancient origination scenario (*Figure 1A*), long-term vertical transmission of Coronaviridae within mammalian lineages could lead to events of mammal-coronavirus cospeciations. Coronaviruses' diversification would then be modulated by both cospeciations and horizontal host switches from one mammalian lineage to another (*Wertheim et al., 2013*; *Shi et al., 2018*). The most recent common ancestor of coronaviruses could even have infected the most recent common ancestor of mammals and birds (*Wertheim et al., 2013*). Under the recent origination scenario (*Figure 1B*), codiversification with hosts is virtually impossible, and coronaviruses' diversification would then be largely dominated by recent host switches. Expectations for the output of reconciliation and biogeographic analyses under these different scenarios, as well as a scenario of random associations, are explained in *Figure 1*. We identify the likely origination of coronaviruses in the mammalian tree, quantify the frequency of cospeciation and host-switching events, and locate these host switches, therefore identifying 'reservoirs' of Coronaviridae and potential transmission routes across mammals.

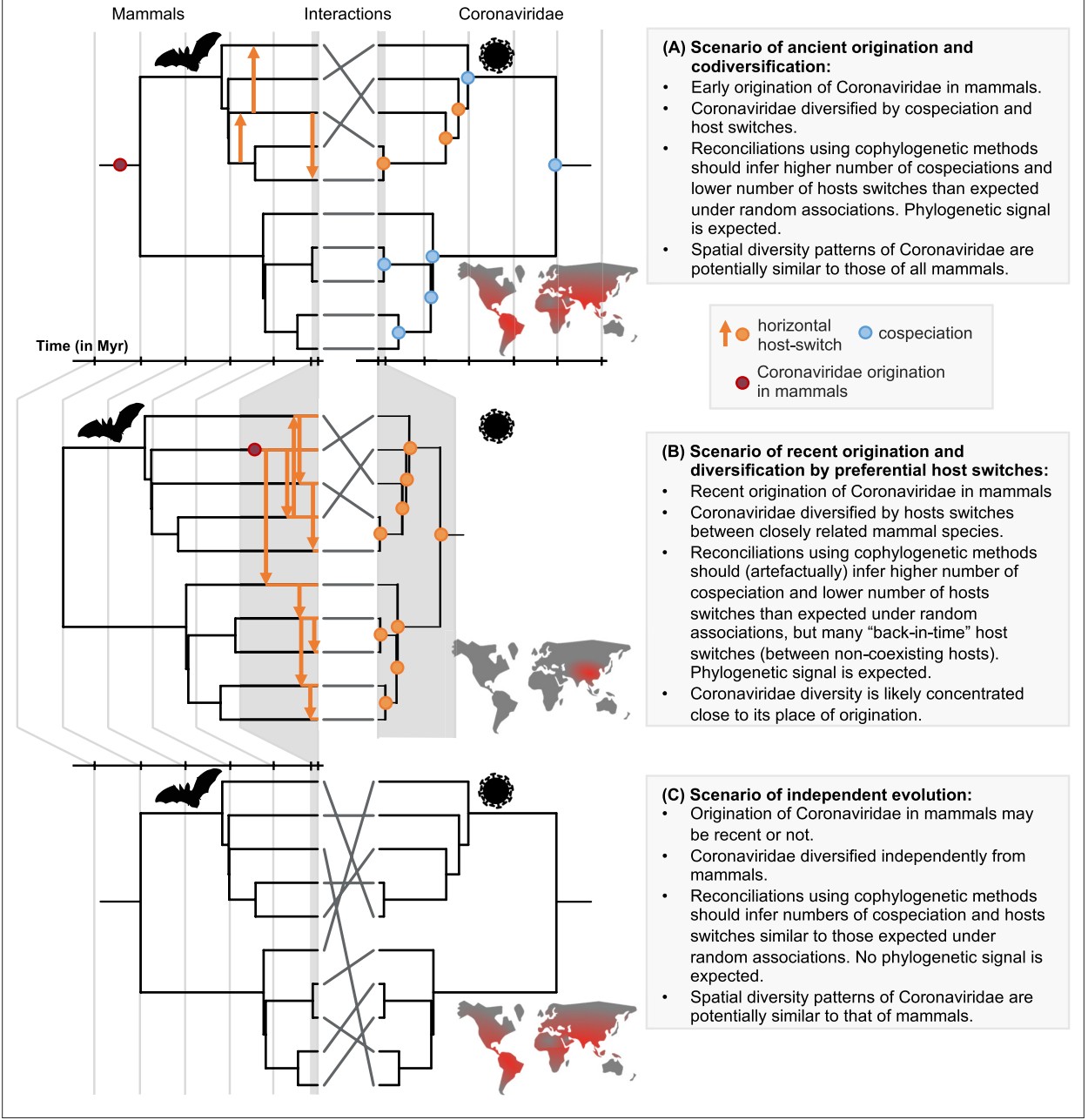

**Figure 1.** A framework for testing scenarios of virus-host evolution, illustrated with the example of Coronaviridae and their mammalian hosts. In (**A**), a scenario of ancient origination and codiversification; in (**B**) a scenario of recent origination and diversification by preferential host switches; and in (**C**) a scenario of independent evolution. For each scenario, we indicate the associated predictions in the grey boxes. Contrary to scenario C, both scenarios A and B are expected to generate a cophylogenetic signal, *i.e.* closely-related coronaviruses tend to infect closely-related mammals, resulting in significant reconciliations when using topology-based probabilistic cophylogenetic methods, such as the undated version of ALE, Jane, or eMPRess. However, we expect scenario B to be distinguishable from scenario A in terms of the time consistency of host-switching events. Under scenario B, cophylogenetic methods wrongly estimate a combination of cospeciations and 'back-in-time' host switches (see Materials and methods and Results). We also expect different biogeographic patterns under the different scenarios, as illustrated by the maps, where the color gradient represents diversity levels (red: high diversity, grey: low diversity).

# Results
## Mammal-coronavirus associations

By screening the 46 sOTUs of Coronaviridae identified by *Edgar et al., 2022* in public databases, we found 35 that were associated with mammalian hosts. Our trees of these 35 sOTUs support a

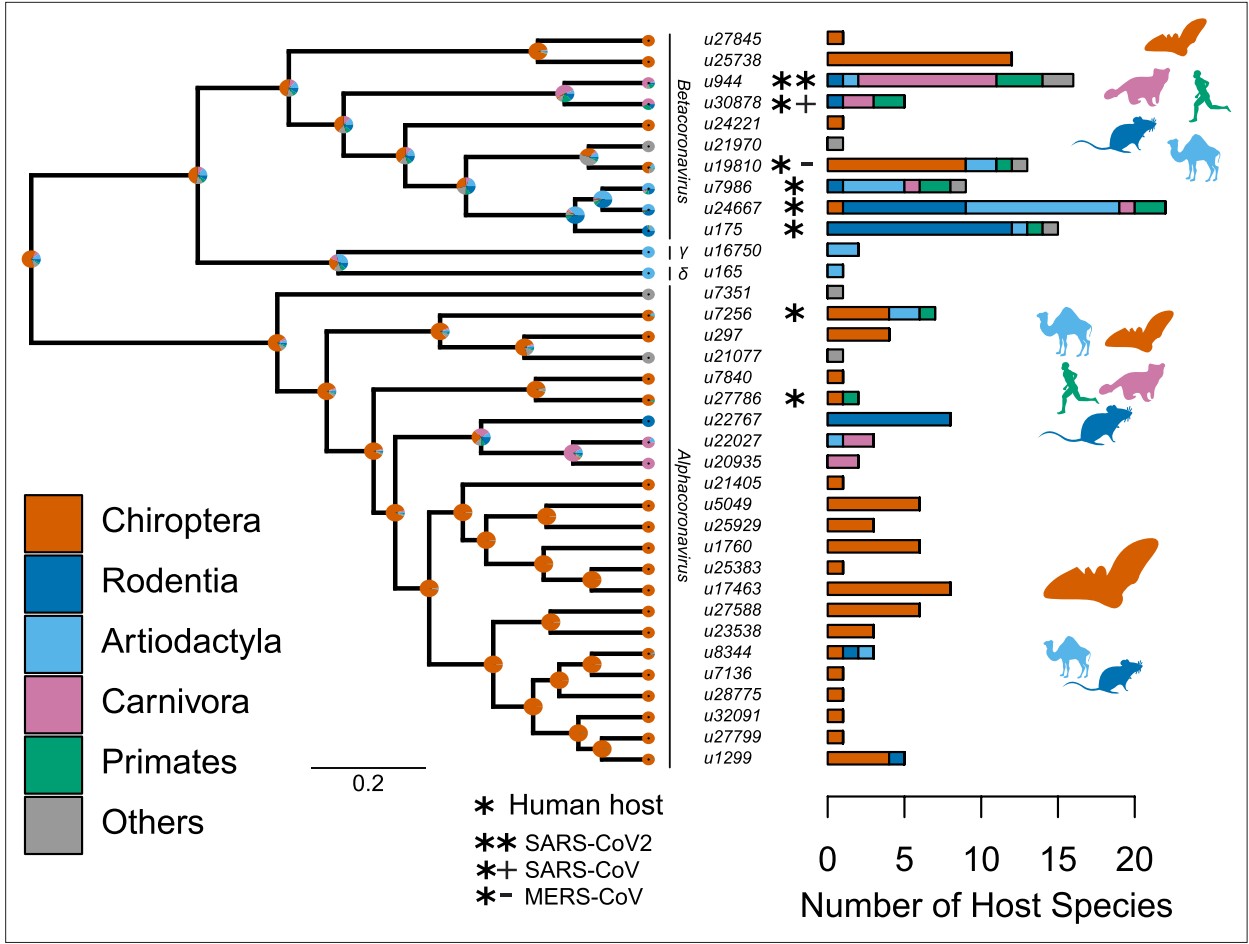

**Figure 2.** Species-level relationships among coronaviruses and their associated mammalian hosts. The Maximum Clade Credibility phylogenetic tree of coronaviruses, reconstructed with BEAST2 based on 150-aa palmprint amino acid sequences of the RdRp gene, is shown on the left. sOTUs of Coronaviridae followed the definition of the Serratus project. The branching order of four genera of coronaviruses, Beta, Gamma, Delta, and Alphacoronaviruses, is shown. Bar scale is in units of aa substitution. On the right, a barplot gives the number of total mammalian host species and the number of host species by main mammalian order. Ancestral states on the left were obtained for illustrative purposes with the make.simmap function of the phytools R package (**Revell, 2012**). Mammal silhouettes taken from open-to-use sources in https://www.phylopic.org, detailed credits given in **Supplementary file 1h**.

The online version of this article includes the following figure supplement(s) for figure 2:

**Figure supplement 1.** Phylogenetic relationships among coronaviruses sOTUs.

**Figure supplement 2.** Mammalian hosts of coronaviruses are shown within the full mammalian tree.

**Figure supplement 3.** The association between coronaviruses and their mammalian hosts.

well-defined split between alphacoronaviruses and the other genera, regardless of the phylogenetic method used (**Figure 2**; **Figure 2—figure supplement 1**). Overall, alphacoronaviruses form a mono-phyletic clade, delta- and gammacoronoviruses form sister clades, with the main uncertainty being on the placement of their ancestor in relation to betacoronaviruses (i.e. as a sister to a monophyletic Beta-clade **Figure 2**) or within the Beta-clade (**Figure 2—figure supplement 1**).

We found that mammalian hosts of coronaviruses belong to 31 families and 10 orders of mammals, and are widely distributed throughout the mammalian phylogeny (**Figure 2**, **Figure 2—figure supplement 2**). Most mammalian hosts are bats (Chiroptera - 55 species), followed by rodents (Rodentia - 22 species), artiodactyls (Artiodactyla - 15 species), carnivores (Carnivora - 11 species), and primates (Primates - 5 species). Five other orders have at least one representative species: Eulipotyphla (**Leao et al., 2022**), Lagomorpha (**Alekseev et al., 2008**), Perissodactyla (**Alekseev et al., 2008**), Pholidota (**Alekseev et al., 2008**), and Sirenia (**Alekseev et al., 2008**). The number of mammalian hosts per coronavirus' sOTU varies across the Coronaviridae tree, ranging from 1 to 22 species, with an

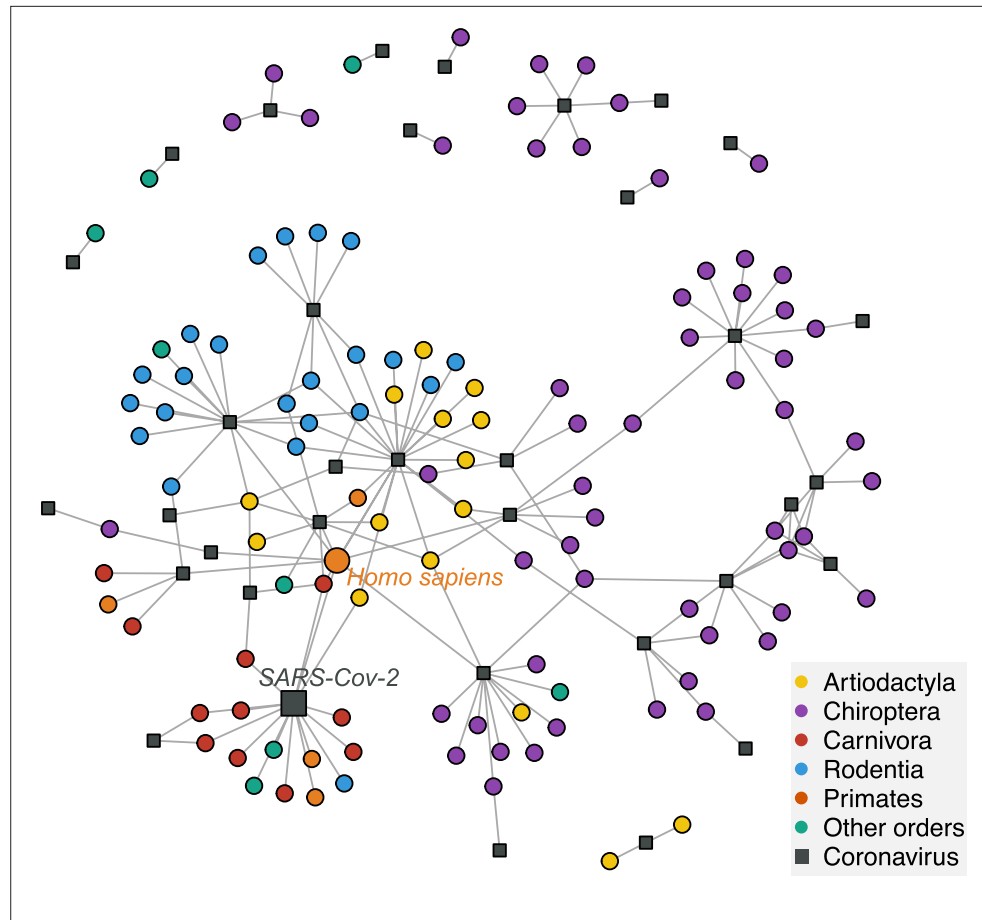

**Figure 3.** A network visualization of mammal-coronavirus interactions reveals the presence of phylogenetic signal, the isolation of bats, and the centrality of humans. Species-level network representation of the interactions between mammal species and coronavirus sOTUs. Colored round nodes represent mammal species (colors indicate the mammalian order) and grey squared nodes correspond to coronavirus sOTUs. The position of the nodes reflects their similarity in interaction partners, i.e. the tendency of clustering of mammals belonging to the same order can be interpreted as the presence of phylogenetic signal in species interactions. Humans and SARS-Cov-2 are presented using bigger nodes. The plot was obtained using the Fruchterman-Reingold layout algorithm from the igraph R-package.

The online version of this article includes the following figure supplement(s) for figure 3:

**Figure supplement 1.** Phylogenetic signal in the association between coronaviruses and their mammalian hosts.

---

average of 4.94 (*Figure 2*). Of the 35 sOTUs, 23 are found in at least one bat species and 17, mostly in alphacoronaviruses, are found exclusively in bats (*Figure 2*). Eight sOTUs are found in humans, six of which, including the three pathogenetic sOTUs, are betacoronaviruses. Betacoronaviruses infect a larger average number of hosts and a larger diversity of non-bat species than alphacoronaviruses. Twenty-two coronaviruses occur in more than one species; of those, 11 are found in multiple orders (*Figure 2*; *Figure 2—figure supplement 3*) and 11 in multiple species of a single order (*Figure 2—figure supplement 3*). A simple visualization of the mammal-coronavirus interaction network indicates that mammals from the same orders tend to be infected by the same coronavirus' sOTUs, that bats tend to host a specific set of coronaviruses, and the centrality of humans in the network (*Figure 3*).

## Phylogenetic signal in coronavirus infections

We first tested whether closely-related coronaviruses tend to infect closely related mammals. A negative answer to this question would suggest that the diversification of Coronaviridae is independent of mammalian history, excluding the scenarios of codiversification or diversification per preferential host switches (*Figure 1*). To the contrary, we found a significant phylogenetic signal for the overall

association between coronaviruses and mammals (Mantel test: $r$=0.38; p=0.0001) and vice-versa ($r$=0.29; p=0.0001), after accounting for the confounding phylogenetic signal in the number of partners (*Perez-Lamarque et al., 2022b*). Mantel tests across sub-clades of both phylogenies revealed that this overall phylogenetic signal is linked to phylogenetic signal in the deep nodes of the Coronaviridae and mammal phylogenies rather than at shallow phylogenetic scales (*Figure 3—figure supplement 1*), consistent with the order-level pattern observed in the mammal-coronavirus interaction network (*Figure 3*). This pattern could arise from ancient codiversification followed by un-preferential host switches, or from recent host switches preferentially occurring between hosts from the same high-level taxonomic grouping (such as mammalian orders). We also found that closely related coronaviruses tend to infect a similar number of hosts ($r$=0.29; p=0.002), while closely related mammals do not tend to host a similar number of distinct coronaviruses ($r$=0.04, p=0.1), suggesting that coronaviruses' specificity towards hosts is evolutionarily conserved while hosts' specificity to coronaviruses is not.

## Diversification dynamics of coronaviruses

To further investigate the hypotheses of ancient codiversification versus recent host switches, we used a probabilistic cophylogenetic model, the amalgamated likelihood estimation (ALE – *Szöllősi et al., 2013*), that reconciles the host and symbiont phylogenies using events of cospeciations, host switches, duplications, or losses, while accounting for phylogenetic uncertainty in the symbiont phylogenies and undersampling of the host species (*Perez-Lamarque and Morlon, 2022*; *Szöllősi et al., 2013*; *Szöllosi et al., 2013*). The main version of ALE we used is an 'undated' version that accounts for topology but not branch lengths, as the dated version did not perform well on our data (see Materials and methods). Time-inconsistent host switches are thus allowed during the reconciliation. If the scenario of ancient diversification holds, we expect to find reconciliations requiring more cospeciations and fewer host switches than expected under a scenario of independent evolution (hereafter referred to as 'significant reconciliation'), and few time-inconsistent switches (*Figure 1A*). Under the alternative scenario of recent origination and diversification by preferential host switches, we also expect to infer a significant reconciliation, but with many time-inconsistent switches, as the algorithm tends to explain the cophylogenetic signal in the interactions by cospeciation events (*Figure 1B*). We indeed found a significant reconciliation between the Coronaviridae and the mammalian trees, confirming the non-independence of their evolution, which we evaluated by randomly shuffling mammal species across the full tree or within biogeographic regions (*Figure 4—figure supplement 1*). ALE reconciliations inferred average numbers of 145 cospeciations, 65 losses, 0 duplication, and 92 host switches. Without investigating the time-consistency of the host switches, we would conclude that there are almost 1.5 more diversification events of Coronaviridae that are related to ancient codiversification rather than host switches. However, on average 20% of the inferred host switches are time-inconsistent, including 'back-in-time' host switches of >50 Myr (*Figure 4—figure supplement 2*). Similar results were observed when accounting for the uncertainty in the branching time estimates (*Figure 4—figure supplement 2*), which suggests instead that extant Coronaviridae originated recently and diversified by frequent preferential host switches.

The majority (64%) of the reconciliations found an origination of coronaviruses within bats, in particular within the Pteropodidae family (*Figure 4A–C*). By comparison, only 19% found an origination in rodents, 6% in artiodactyls, and 2% in carnivores. In addition, we no longer found an origination in bats when randomly shuffling the dataset (*Figure 4—figure supplements 3 and 4*), suggesting that this result is not artifactual. We checked the interpretation of our results by simulating the two scenarios of (i) ancient origination in the ancestors of bats followed by codiversification and (ii) recent origination in an extant bat species and a subsequent diversification by preferential host switches. On the first set of simulations, ALE correctly inferred an origination in bats and very few time-inconsistent switches (2% +/- 2%), which seems to be the basal expected proportion of time-inconsistent switches under a scenario of codiversification (*Figure 4—figure supplement 5*). On the second set, ALE correctly inferred an origination in bats, although with lower confidence, and a large fraction (~20%) of time-inconsistent host switches, similar to what we observed for Coronaviridae. These results therefore indicate a scenario of recent origination of coronaviruses in bats followed by diversification by preferential host switches.

To investigate this scenario in more detail, we gradually applied a tree transformation to the mammalian phylogeny, which excludes the possibility of an ancient origination happening earlier

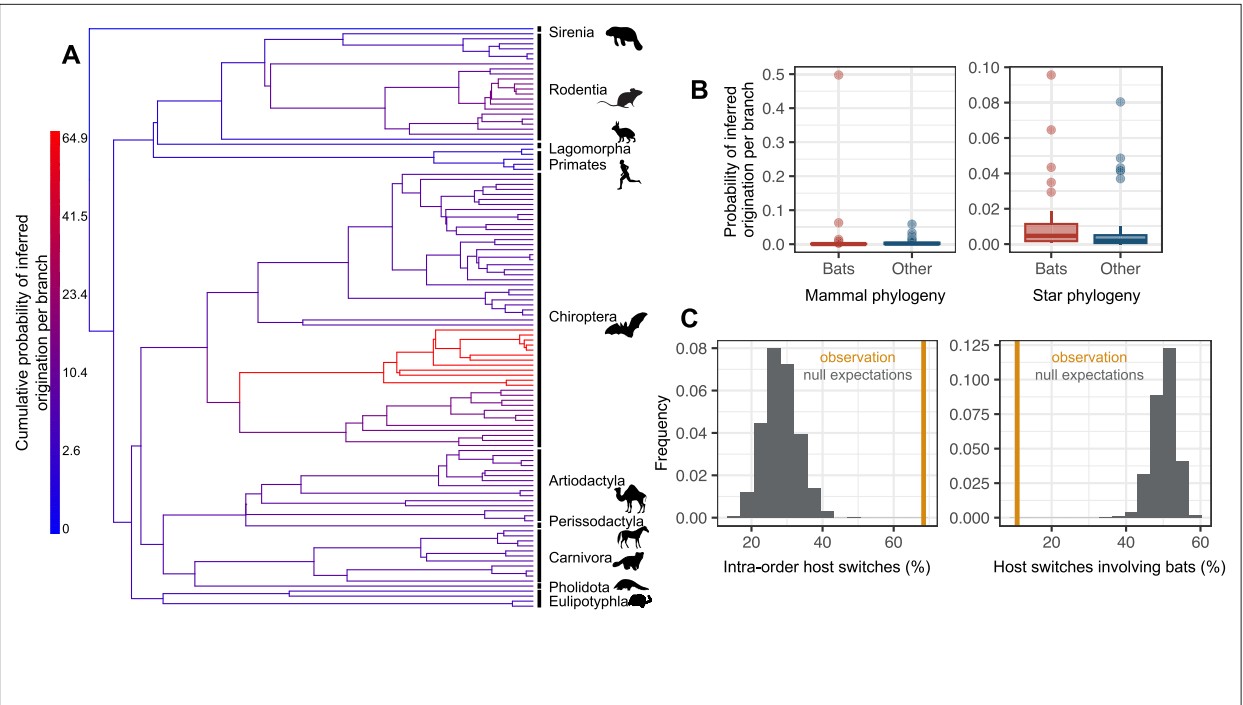

**Figure 4.** The origination of coronaviruses in mammals is estimated among bats, which tend to form a closed reservoir. (**A**) Phylogenetic tree of the mammals with branches colored as the percentage of ALE reconciliations which inferred this branch or its ancestral lineages as the origination of coronaviruses in mammals. Red branches are likely originations, whereas blue branches are unlikely. (**B**) Boxplots recapitulating the probability of inferred origination per branch in bats *versus* other mammal orders, with ALE applied on the original mammal tree (left panel) or on the mammal tree transformed into a star phylogeny (right panel), therefore assuming an origination in extant species. (**C**) Distributions of the percentages of host switches occurring within mammalian orders (left panel) and between-orders involving bats (right panel). Observed values (in orange) are compared to null expectations if host switches were happening at random (in grey). Mammal silhouettes taken from open-to-use sources in https://www.phylopic.org/, detailed credits given in *Supplementary file 1h*.

The online version of this article includes the following figure supplement(s) for figure 4:

**Figure supplement 1.** ALE inferred significant reconciliations.

**Figure supplement 2.** Time-inconsistent host-switches.

**Figure supplement 3.** The origination of coronaviruses in mammals is not estimated among bats anymore when shuffling the dataset.

**Figure supplement 4.** The origination of coronaviruses in mammals is estimated among bats.

**Figure supplement 5.** Validation of the interpretation of our results on the mammalian phylogeny using simulations of codiversification (left) or diversification by preferential host switches (right).

**Figure supplement 6.** Validation of the interpretation of our results on the star phylogeny using simulations of diversification by preferential host switches.

**Figure supplement 7.** Simulating a scenario of origination in rodents followed by a diversification by preferential host switches with higher diversification of coronaviruses within bats did not generate a spurious origination in bats.

**Figure supplement 8.** Evidence of preferential host switches in coronaviruses.

**Figure supplement 9.** Host switches are less likely than expected by chance between bats (Chiroptera) and Artiodactyla or Rodentia.

**Figure supplement 10.** The frequency of host switches seems to vary according to the coronavirus lineages.

than a given time. We found that we had to impose a very recent time of origination (younger than 5 Myr) to obtain few time-inconsistent switches (*Supplementary file 1b*). We thus carried out our follow-up analyses with a mammals' tree transformation (star phylogeny) that assumes an origination in an extant mammalian lineage, such that coronavirus diversification is explained entirely by host switches between extant mammalian species. Simulations validated this approach in terms of properly inferring originations and identifying preferential host switches (*Figure 4—figure supplement 6*). Applied to the data, the approach inferred a high probability of origination in bats (56%, *Figure 4B-C*, *Figure 4—figure supplement 4*), far more likely than in other mammalian orders (artiodactyls: 18%,

rodents: 7%, carnivores: 5%). Simulations confirmed that our results did not spuriously arise because of higher coronaviruses diversification in bats compared to other mammalian orders (*Figure 4—figure supplement 7*).

Following the origination in bats, our approach based on the star phylogeny inferred a scenario of diversification by preferential host switches: 68% of the inferred host switches happened within mammal orders (*Figure 4D*, *Figure 4—figure supplement 8*), whereas we would expect on average only 28% of within-order host switches if happening at random. We also inferred more-than-expected host switches between closely related mammal orders (e.g. between Artiodactyla and Perissodactyla) and between the order containing humans (Primates) and those of their domesticated animals, such as Artiodactyla and Carnivora (*Figure 4—figure supplement 8*, *Supplementary file 1c*). In contrast, host switches were five times less numerous than expected by chance between bats and other orders (10.7%, against 50.2% on average if host switches were randomly distributed, *Figure 4D*), in particular Artiodactyla and Rodentia (*Figure 4—figure supplement 9*, *Supplementary file 1c*). When occurring, host switches from bats often occurred toward humans (1.9 host switches per reconciliation on average) or toward urban-living and/or domesticated animals, such as rats, camels, or pigs (>1 host switch on average; *Supplementary file 1d*). Host switches to humans occurred mostly from domesticated mammals (camels, pigs, dogs), the house shrew and the house mouse, then followed by Asian palm civets, and lastly by bats and other rodents (*Supplementary file 1e*). Finally, we found that some sOTUs, in particular from betacoronaviruses (e.g. u24667 and u175, both with humans among their hosts), have experienced frequent host switches, whereas others have not (e.g. u165, which is restricted to pigs). In particular, u944 (SARS-Cov-2) has experienced an intermediate number of host switches compared to other coronaviruses (*Figure 4—figure supplement 10*).

We also separately investigated the diversification dynamics of the two main clades of coronaviruses: the alpha- and betacoronaviruses. The undated version of ALE inferred, in alphacoronaviruses, 64 cospeciations, 0 duplication, 35 host switches, and 25 losses, as well as a recent origination in bats (69% of the reconciliations) and frequent intra-order host switches (80%). In contrast, in betacoronaviruses, a majority of the reconciliations (76%) had an origination in mammalian orders other than bats, including rodents (25%), Artiodactyla (18%), or Carnivora (13%), and involved inter-order host switches (61%). These results suggest that the ancestral coronavirus originated in bats, gave rise to alphacoronaviruses in bats, and switched to a different non-bat host where it evolved into betacoronaviruses. With 80 cospeciations, 0 duplication, 50 switches, and 36 losses, the fraction of host switches relative to cospeciation events is higher in betacoranaviruses (0.63) than in alphacoronaviruses (0.54). In both clades, host switches from bats still occurred preferentially toward humans or domesticated mammals, and >15% of the switches were time inconsistent. The dated version of ALE that forces host switches to be time-consistent failed to output a reconciliation for betacoronaviruses; in alphacoronaviruses, we obtained significant reconciliations with more 'mismatch' events (*R Development Core Team, 2018*, including 38 host switches and 29 losses) than cospeciation events (*Bouckaert et al., 2014*), suggesting cophylogenetic signal without phylogenetic congruence (*Supplementary file 1a*; *Figure 1*; *Perez-Lamarque and Morlon, 2024*). While the latter result would be inconclusive, when it is combined with the numerous time-inconsistent switches found with the undated ALE version, it suggests that a scenario of codiversification is very unlikely.

## Sensitivity analyses

We carried a series of sensitivity analyses to assess the robustness of our analyses to potential biases or issues, summarized in *Supplementary file 1f*.

We found qualitatively similar results when applying ALE on different sub-parts of the palmprint region, suggesting that the potential occurrence of recombination does not bias our conclusions (*Supplementary file 1g*). The percentage of originations inferred to occur in bats decreased in the analyses on the first sub-part, probably because using such a short fragment (75 aa-long) does not allow robust reconciliations. We also obtained consistent results using a reconciliation method based on maximum parsimony (eMPRess) instead of maximum likelihood (ALE). Whatever the costs that we set for the different reconciliation events, eMPRess estimated significant reconciliations (p<0.01). For instance, when favoring host switches, we inferred a recent origination in bats in 54% of the reconciliations and observed on average 32 cospeciations (s.d. +/-3), 2 losses (s.d. +/-1), 0.1 duplication (s.d. +/-0.3), and 140 host switches (s.d. +/-3) including several 'back-in-time' host switches

of >30 Myr. eMPRess therefore also supports a scenario of recent origination in bats and diversification by preferential host switches (*Figure 1B*). Without investigating the time-consistency of the host switches, we would have wrongly concluded that almost one fourth of the diversification of Coronaviridae is related to ancient cospeciation events.

Our understanding of extant mammal-coronavirus interactions is incomplete and subject to taxonomic and geographic biases (*Poisot et al., 2023*). To address potential biases, we investigated the potential effect of sampling biases on our results by sub-sampling our dataset. First, we tested the effect of unequal sampling effort within screened mammal species (e.g. humans and domesticated animals have been more extensively sampled than wild animals). When randomly subsampling only three Genbank accession codes per host species, we still found frequent originations in bats (59%+/-s.d. 5%), preferential host switches (48%+/-s.d. 5%), and frequent transfers from bats to humans (in 90% of the subsampled dataset) or domesticated animals (e.g. *Sus scrofa* and *Camelus dromedarius* in 100% and 52% of the subsampled dataset, respectively). Second, we tested the effect of unequal sampling effort across the mammal tree of life. When randomly subsampling up to 10 species per mammalian order, we observed an average decrease in the probability of origination in bats (from 56% in the original dataset to 37%+/-s.d. 10%), yet this scenario remained much more likely than an origination in other mammalian orders, such as artiodactyls (14%+/-s.d. 4%), rodents (11%+/-s.d. 3%), or carnivores (10%+/-s.d. 2%). Transfers from bats still occurred more frequently toward humans or domesticated mammals (*Camelus dromedarius*, *Rattus norvegicus*, or *Sus scrofa*): along with *Sorex*

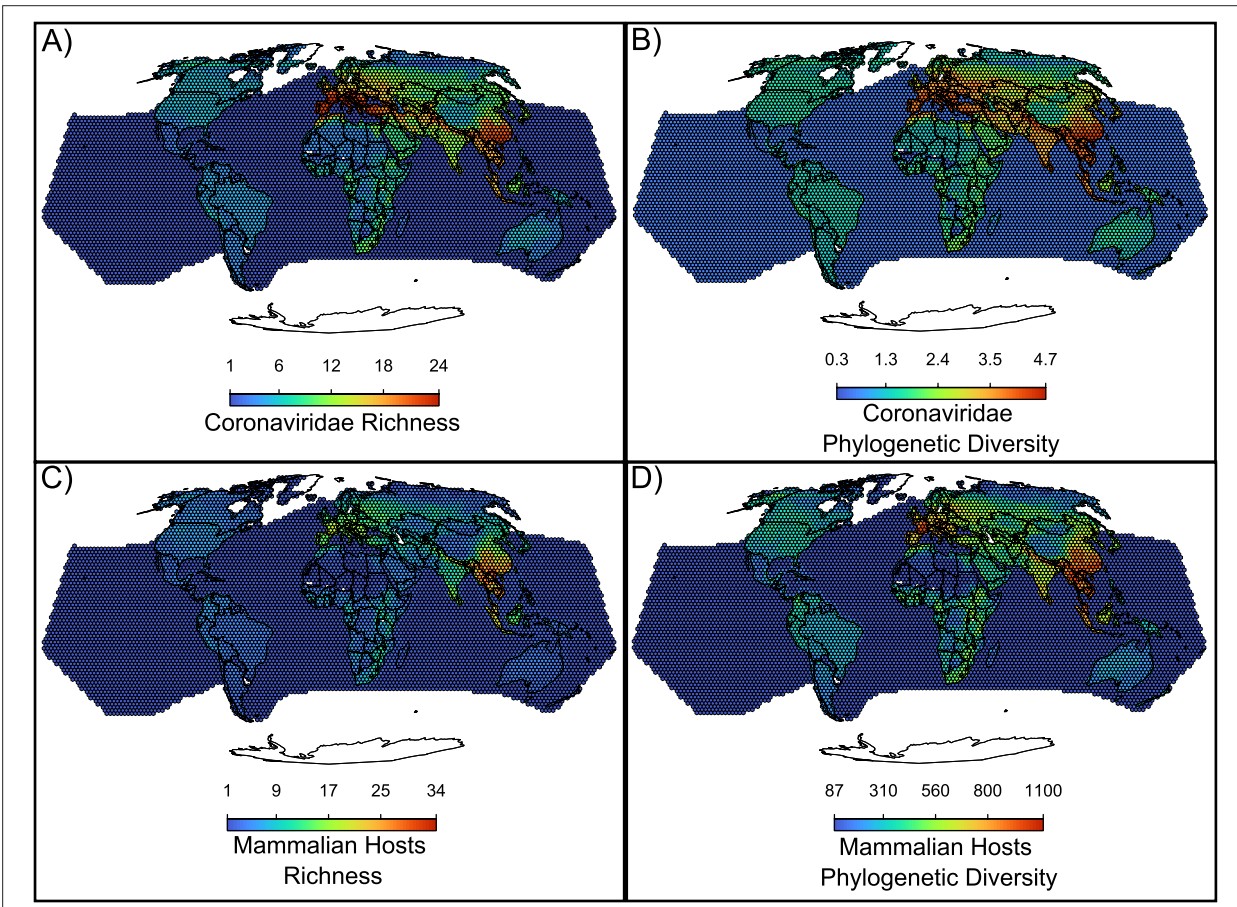

**Figure 5.** Maps of the diversity of coronaviruses and their mammal hosts. In (**A**), the richness of species of coronaviruses; geographic range maps of coronaviruses were constructed after applying the host-filling method on the geographic range maps of mammalian hosts of coronaviruses. In (**B**), *Faith, 1992* phylogenetic diversity of coronaviruses, calculated using the phylogenetic tree of coronaviruses (see main text). In (**C**) and (**D**), the richness and phylogenetic diversity of mammal hosts of coronaviruses, respectively. All maps are on the Mollweide projection.

The online version of this article includes the following figure supplement(s) for figure 5:

**Figure supplement 1.** Maps of the diversity of alpha and betacoronaviruses.

*araneus*, these species were in the top 5 of the species receiving a coronavirus from bats*ffing. Overall, these sensitivity analyses* indicate that the likely origination in bats and the frequent transfers from bats to humans or domesticated animals is not artifactually driven by the high number of bats species in the dataset nor the enhanced monitoring of coronaviruses in humans or domesticated animals.

## Geographical distribution of coronaviruses

An additional piece of evidence for a recent origination scenario comes from the geographical distribution of coronaviruses, with a hotspot of diversity in Eurasia that has not colonized the whole world (*Figure 5A*, *Figure 1B*). The coronavirus' hotspot is more strongly influenced by the diversity of alphacoronaviruses than of betacoronaviruses (*Figure 5—figure supplement 1*). The higher host switches rates and broader host range of betacoronaviruses is reflected in a more widespread geographic distribution, with less pronounced hotspots when compared to alphacoronaviruses (*Figure 5—figure supplement 1*). Mammalian hosts of coronaviruses have a hotspot of species diversity concentrated in East Asia (*Figure 5C*). The richness of coronaviruses presents a similar pattern, but with two comparable hotspots of species diversity in East Asia and Southern Europe (*Figure 5A*), suggesting that the European hotspot is composed by fewer host species, together carrying as diverse a set of coronaviruses as the Asian hotspot. Other regions with a relatively high richness of coronaviruses and their hosts include parts of the African continent. The Americas and Australia have relatively low richness of coronaviruses and their hosts. Phylogenetic diversity of both hosts and coronaviruses (*Figure 5B and D*) depict a similar pattern but with phylogenetic diversity more evenly distributed across most world regions, including the Americas.

## Discussion
### Inferring the coronaviridae evolutionary history

Together, our results suggest that the common ancestor of extant mammalian coronaviruses originated recently in a bat species, and that coronaviruses diversification occurred via preferential host switches rather than through codiversification with mammals. Although we cannot unequivocally date the timing of origination of coronaviruses in mammals, we demonstrate that Coronaviridae is a highly dynamic clade in which diversification operates through host switches at a much faster pace than that of their hosts. sOTUs are rapidly replaced by newly generated ones, with little role for codiversification with the hosts. The high diversity and endemicity of coronaviruses among bats have led others to anticipate that bats might be implicated in the origin of coronaviruses (*De Groot, 2022*; *Corman et al., 2018*; *Woo et al., 2012*; *Vijaykrishna et al., 2007*), although definitive proof was lacking. We provided evidence for that hypothesis using a probabilistic cophylogenetic model and accounting for the entire known diversity of coronaviruses across mammals. Independent evidence for coronavirus recent host switches among different species exists in the literature (*Dudas et al., 2018*; *Wei et al., 2021*). The envisioned scenario suggests a timing of origination for extant Coronaviridae that is much more recent than the hundreds of millions of years ago suggested by *Wertheim et al., 2013*. This is not surprising given the difficulties in estimating divergence times and inferring branch lengths for viral phylogenies (*Wertheim et al., 2013*; *Wertheim and Kosakovsky Pond, 2011*; *Duchêne et al., 2014*), and provided that the dating of *Wertheim et al., 2013* relied on a substitution rate estimated from data with limited temporal signal (~50 serially sampled contemporary sequences of a short gene fragment, *Woo et al., 2012*).

Our results contradict previous suggestions that codiversification with vertebrate hosts played an important role in Coronaviridae diversification (*Wertheim et al., 2013*; *Hayman and Knox, 2021*; *Shi et al., 2018*). They also suggest that previously reported cases of long-term codiversification in vertebrate RNA viruses have been largely over-estimated, as many of them may instead be cases of diversification by host switches occurring preferentially among closely related hosts (*Geoghegan et al., 2017*). Indeed, these two scenarios both generate cophylogenetic signal in host-symbionts associations, such that cophylogenetic signal alone is not evidence for long-term codiversification (*de Vienne et al., 2013*). In addition, under a scenario of recent origination and preferential host switches, event-based cophylogenetic methods tend to artifactually favor biologically unrealistic scenarios with codiversification and back-in-time host switches, as we have shown here. As the time-consistency of host switches is typically not investigated, this has remained unnoticed, and evidence

for codiversification has been taken for real. Ideally, cophylogenetic reconciliation methods would not allow such time-inconsistent host switches. However, imposing time constraints in methods based on parsimony is NP-hard (*Santichaivekin et al., 2021*), and the 'dated' version of ALE is not well adapted when recent host switch events dominate evolutionary history. We have found two ways to get around the problem, by interpreting time-inconsistent host switches as evidence for recent preferential host switches, and by gradually transforming the host tree to avoid large back-in-time switches, however future efforts should focus on developing time-consistent cophylogenetic methods. This would allow more robust and precise inferences of host-virus (and more generally host-symbiont) evolutionary history. We have shown by simulation that contrasting scenarios, such as codiversification, or diversification by preferential host switching after originating within or outside bats, leave distinct signatures in the data. This suggests that deep neural networks trained on simulated data, which have already shown their performance in phylodynamics (*Lajaaiti et al., 2023*; *Thompson et al., 2023*; *Voznica et al., 2022*; *Lambert et al., 2023*), could also be useful for the analysis of cophylogenetic data.

## Host switching dynamics in coronaviridae

During their evolution, coronavirus' host switches occurred more frequently within than between mammalian orders. This suggests that mammalian characteristics shared between relatives (e.g. genetic, behavioral, ecological), and the frequency of encounters among hosts play important roles in determining coronavirus' host switches. Additionally, between-order host switches occurred more frequently among non-flying mammals and among orders containing humans and urban and domesticated mammals, suggesting that contact frequency alone is likely a key characteristic in host switches. Accordingly, amongst the most-likely host switches towards humans were those coming from mammals suspected to be involved in the transfer of specific coronavirus sOTUs likely through contact, for instance, camels in the case of MERS-CoV (*Dudas et al., 2018*), Asian palm civets with SARS-CoV (*Guan et al., 2003*; *Chinese SARS Molecular Epidemiology Consortium, 2004*) and the house mouse with SARS-CoV-2 (*Wei et al., 2021*). Importantly, we found that host switches from bats to other mammalian species were rare during the evolutionary history of Coronaviridae, even though coronaviruses originated and are more diverse within bats. These pieces of evidences suggest that bats are a closed reservoir of the Coronaviridae diversity, as also suggested by their relative isolation in the mammal-coronaviruses interaction network (*Figure 3*).

Spillovers from bats to non-bat species, when they occurred, were found more likely to be towards humans than to any other mammalian species, suggesting humans may have acted as evolutionary intermediate hosts amongst mammals, in line with their centrality in the mammal-coronaviruses interaction network (*Figure 3*). From an ecological perspective, the large abundance and widespread geographic distribution of humans, together with our habits of forcing contact with other species, including bats, make it unsurprising that humans, among all mammal species, have acted as intermediate hosts of ancestral forms of coronaviruses. Interestingly, for some individual species of coronaviruses, such as the SARS-CoV2 and other SARS-like coronaviruses, the dominant hypothesized scenario is that precursor forms spread from a bat to another intermediate mammalian host before infecting humans (*Corman et al., 2018*; *Tan et al., 2022*). Our molecular marker lacks the intra-I resolution necessary to make species-level predictions, but our results suggest that more ancient coronaviruses host switches may have occurred in the other direction: from bats to humans to non-bat mammals. The large spillover of SARS-CoV2 from humans to wild mammalian lineages has now been well documented (*Tan et al., 2022*; *Goldberg, 2023*) and tend to confirm our results of humans as the intermediate host. Many human activities lend credit to the human-as-evolutionarily-intermediate-host-hypothesis, including human excursions to bat caves (*Furey and Racey, 2016*), hunting (*Mildenstein et al., 2016*), and habitat destruction and modification (*Smith and Wang, 2013*), all of which increase the contact between bats and humans and their domesticated animals (*Smith and Wang, 2013*). Conservation of bats' natural habitats, away from human contact, could help avoiding further spreads of coronaviruses among humans.

In agreement with previous studies *De Groot, 2022*; *Mavrodiev et al., 2020*; *Woo et al., 2009*; *Woo et al., 2012*; *Vijaykrishna et al., 2007*,)), we found noticeable differences between Alpha- and Betacoronaviruses. While Alphacoronaviruses likely originated in bats and mostly diversified by switching between bat lineages, betacoronaviruses most likely originated in a non-bat lineage and experienced a majority of their host switches between non-bat mammals. In addition, our findings

suggest that host switching rates are higher in betacoronaviruses than in alphacoronaviruses. These different host switching dynamics are consistent with biogeographical differences: betacoronaviruses tend to be more geographically spread than alphacoronaviruses, which are more restricted to Eurasia. Some of the trends we observed (e.g. the higher host switching rates in betacoronaviruses) are not consistent with previous findings (*Caraballo, 2022*; *Latinne et al., 2020*, however these results are not directly comparable, as studies were conducted at very different scales (across the Americas *Caraballo, 2022*, across China *Latinne et al., 2020*, and here at the global scale).

Insights of past and future host switches are gained from coronavirus geographic distribution. Coronaviruses are found worldwide and their hotspots of diversity are concentrated in East Asia and Southern Europe, where they likely originated. Previous assessments of the diversity of bat hosts of betacoronaviruses suggested similar hotspots but with a distribution of coronaviruses more concentrated in the hotspots (*Becker et al., 2022*; *Anthony et al., 2017*; *Munoz, 2022*) than the more pervasive pattern we found using all mammalian hosts. Moreover, the distribution of coronaviruses is less concentrated in the hotspots when phylogenetic metrics of diversity are included, suggesting that species richness alone is masking the global evolutionary potential of these viruses (*Leopardi et al., 2018*). Coronaviruses' likely recent origination in bats, high within-order transmission rates, and their capacity to switch between mammal orders in some cases suggest the potential for future fast spreading and increase in the number of species across most world regions. Among alphacoronaviruses, the spread is more likely to remain concentrated within bats, while betacoronaviruses have a higher potential for among-orders spreading and infection of new mammalian hosts. The betacoronaviruses lineages already detected in humans are host generalists with high transmission rates, suggesting that continued monitoring may be wise in mitigating potential future pandemics.

## Limitations and perspectives

A few important limitations of our analyses deserve to be mentioned. First, we used a short marker gene to reconstruct the Coronaviridae phylogeny. However, the RdRp palmprint marker we used is very conserved and therefore routinely used for delimiting operational taxonomic units in RNA viruses and reconstructing their evolutionary history (*Edgar et al., 2022*; *Babaian and Edgar, 2022*). Our tree has an identical genus-level topology compared with trees constructed using other genome regions, such as the nucleocapsid portion of the coronavirus genome (*Woo et al., 2012*), and very close topologies were also observed using other genome regions such as the spike, envelope, and membrane regions (*Woo et al., 2012*), as well as the entire RdRp region (*De Groot, 2022*; *Zmasek et al., 2022*; *Woo et al., 2012*): it differs only on the placement of Gammacoronavirus and Deltacoronavirus in relation to the others, but it is consistent in the monophyly of both Alphacoronavirus and Betacoronavirus and recovers the sister-clade relationship between Gamma and Delta. In addition, because the palmprint region is a conserved region, we could not reconstruct the recent evolutionary history of coronaviruses (i.e. the within sOTU transmission dynamic). Combining the palmprint region with a fast-evolving region(s) would enable more precise estimates of the recent routes of coronaviruses' transmission, including that of SARS-CoV-2. More generally, characterizing multiple genetic regions with different evolutionary rates across Coronaviridae would allow us to more precisely elucidate the timescale of the evolutionary history of coronaviruses alongside their hosts.

Second, recombination is an important mechanism of viral evolution (*Pérez-Losada et al., 2015*), and approaches more adequately designed to investigate the role of recombination are needed. The fact that different subparts of the palmprint region lead to similar results indicates that recombination acting on the palmprint region is unlikely to bias our conclusions. However, looking at other genomic regions would allow gaining a more complete understanding of the role of recombination in coronavirus evolution.

Third, the record of associations between coronaviruses and mammals is necessarily incomplete (not all mammal species have been screened for coronaviruses) and likely biased towards bats and mammals that are in contacts with humans. This a common bias when studying mammal-virus associations that may cause various issues (*Poisot et al., 2023*). However, our sub-sampling analyses suggest that our main results (recent origination in bats and frequent transfer from human-associated mammals to humans) are not artifactually driven by sampling biases. ALE explicitly accounts for undersampling by assuming that host switches involve unsampled intermediate hosts, which may explain the robustness of our findings to sampling biases.

## Concluding remarks

Understanding the evolutionary origins and diversification of viruses is useful for predicting new transmission routes, yet the relative frequencies of virus–host cospeciation versus cross-species transmission in the evolution of vertebrate RNA viruses remains uncertain (*Shi et al., 2018*). We found that coronaviruses originated in bats where they are more diverse nowadays, and later diversified in other mammal orders through preferential host switches. Spillovers from bats were rare but likely human-induced, suggesting humans are the intermediate evolutionary bridge that facilitated the spread of coronaviruses across mammals. Host switches between primates and artiodactyls, perissodactyls, and carnivorans occur frequently. This suggests a potential for the spread of coronaviruses to new mammalian hosts beyond their current prevalence in East Asia and Europe, and raises concerns about the possibility of future pandemics related to coronaviruses. Our results indicate that reducing human-bat contact, for instance through the conservation of bat habitats, could potentially serve as a mitigation strategy. They also suggest that cases of long-term virus–host codiversification, reported on the basis of cophylogenetic tests, have been largely over-estimated.

# Materials and methods

## Operational taxonomic units for coronaviridae

Viral species delimitation is difficult (*Gorbalenya et al., 2020*), and the number of proposed and/or estimated species or strains of Coronaviridae in the literature have varied (e.g. 100 proposed - 3204 estimated *Anthony et al., 2017*; 88 proposed - 204 estimated *Wardeh et al., 2021*). However, an official assessment and classification of viruses are made by the International Committee on the Taxonomy of Viruses (ICTV - *Adams et al., 2017*), by means of its Study Groups (*Gorbalenya et al., 2020*). The Coronaviridae Study Group of the ICTV (*de Groot et al., 2013*) suggests that species delimitation within Coronaviridae should be made considering more than 90% amino acid sequence identity in conserved replicase domains as a criterion to include sequences in the same species (*De Groot, 2022*). Estimates following ICTV suggestions have proposed between 17 and 39 species of coronaviruses (*Gorbalenya et al., 2020*; *Campos and Lourenço-de-Moraes, 2020*; *De Groot, 2022*), and the last (year 2021 v3) Master Species List from the ICTV lists 54 species in the family Coronaviridae (https://ictv.global/msl). Recently, a seminal paper by 38 proposed the use of a taxonomic barcode based on the palmprint region of the RdRp region for the systematic identification and classification of RNA viruses. The use of the RdRp region of the RNA genome is also common in other tree construction attempts for Coronaviridae (*Woo et al., 2012*; *Wertheim et al., 2013*), given that the RdRp is an essential enzyme for replicating the RNA genome (*Venkataraman et al., 2018*), and therefore aligns well with the ICTV proposition.

We therefore used the 46 described species-like Operational Taxonomic Units (sOTUs) for Coronaviridae delimited using 'palmprint' sequences by *Edgar et al., 2022*; *Babaian and Edgar, 2022*. The palmprint is a conserved amino acid (aa) sub-sequence (150 aa in Coronaviridae) of central importance in the viral RdRp (*Edgar et al., 2022*), selected for its homology across the large majority of sequences, allowing estimation of sequence divergence and phylogenetic trees (*Babaian and Edgar, 2022*). sOTUs were identified by *Edgar et al., 2022* after clustering palmprint sequences at 90% amino acid identity; and released through the Serratus project. Despite its relatively short length, trees constructed with this region are topologically equivalent to Coronaviridae trees based on other genes (see Discussion). IWe downloaded the palmprint amino acid sequences of Coronaviridae sOTUs from the Serratus project (https://serratus.io/; *Edgar et al., 2022*) on April 13 of 2022.

## Mammalian hosts of coronaviridae

All 46 sOTUs of Coronaviridae with a full palmprint and associated data in the NCBI database were screened for the identification of its hosts. From those, 35 sOTUs were associated with mammalian hosts and were kept for downstream analyses. Serratus' associated metadata was used to identify GenBank accession codes linked to each sOTU. The complete set of 90,540 associated GenBank accession codes was screened to obtain the host information for each sOTU (on NCBI, Features >source > /host=). All the host species with a full Linnean name were kept as such. Accession codes with hosts leading to a generic level information were further inspected to identify the associated publication and determine the complete species name. Dubious cases or accession codes without publications

had their hosts disregarded. Common names or high-level host information (e.g. host='bats') were generally eliminated except in a few cases where a domesticated species was found to be the host (i.e. host='dog', 'canine' were *Canis lupus*; host='cat', 'feline' were *Felis catus*; host = 'pig', 'piglet', 'newborn piglet', 'sucking piglet', 'porcine', 'swine' were *Sus scrofa*). A final dataset of 116 mammalian hosts associated with the 35 sOTUs was assembled and used in downstream analyses. A matrix with the association between Coronaviridae sOTUs and mammalian species is available in ource data 1.

## Coronaviridae phylogenetic trees

We constructed a Coronaviridae tree using the palmprint amino acid sequence information of the 35 sOTUs. We aligned the amino acid sequences with MAFFT (*Katoh and Standley, 2013*) and trimmed them with trimAl (*Capella-Gutiérrez et al., 2009*). The final alignment contained 150 amino acid positions. We used two main phylogenetic software, BEAST2 (*Bouckaert et al., 2014*), which performs rooting and time calibration and PhyloBayes (*Lartillot and Philippe, 2004*), which generates outputs adapted to the cophylogenetic algorithm we used. We visualized phylogenetic trees using R (*R Development Core Team, 2018*).

In order to run BEAST2, we generated an input file using BEAUti with 35 sOTU sequences and the following parameters: a WAG model with four classes of rates and invariant sites, a birth-death prior, and a relaxed log-normal clock. BEAST2 sampled a posterior distribution of ultrametric trees using Markov chain Monte Carlo (MCMC) with four independent chains each composed of 100,000,000 steps sampled every 10,000 generations. We checked the convergence of the 4 chains using Tracer (*Rambaut et al., 2018*). We used LogCombiner to merge the results setting a 25% burn-in and TreeAnnotator to obtain a Maximum Clade Credibility (MCC) tree with median branch lengths. Using a LG model instead of WAG also gave a consensus tree with a very similar topology.

To further assess the robustness of the BEAST2 tree rooting, we estimated the root position on a 46-sOTU maximum likelihood tree (from the Serratus pr–ject – *Edgar et al., 2022*) assuming a strict molecular clock and an ultrametric tree. We used an ultrametric setting as temporal information from the tip dates (ranging between 1999 and 2022, a negligible difference with respect to the root age of dozens of thousands or even millions of years) was not sufficient to infer the mutation rate (we assessed the temporal signal with TempEst, *Rambaut et al., 2016*). We performed rooting and time-scaling with LSD2 (v2.3, *To et al., 2016*), assuming a tree of unknown scale (e.g. fixing all the tips dates to 1 and the root date to 0) with outlier removal and root search on all branches. LSD2 detected no outliers and positioned the root on the same branch as in the BEAST2 MCC tree (between alpha and betacoronaviruses).

The tree we constructed with PhyloBayes was specifically designed to the application of cophylogenetic methods, which cannot handle the presence of the same symbiont species in multiple host species. Following (*Perez-Lamarque and Morlon, 2022*), we multiplicated the sOTU sequences in cases when a sOTU was present in several mammal species, such that each mammal-coronavirus association is represented by one single sequence (total of 173 sequences). We reconstructed the phylogenetic tree of the 173 sequences with PhyloBayes, run using an LG model, 4 classes of rates, and a chain composed of 4000 steps with a 25% burn-in. To sum up, we built two trees for coronaviruses: a 35-tip tree with BEAST2 (with one tip per sOTU) and a 173-tip tree with PhyloBayes (with multiple tips per sOTU corresponding to the multiple host species associated with each sOTU, such that cophylogenetic methods can run).

## Mammalian phylogenetic tree

We obtained a phylogenetic hypothesis for mammals from the consensus DNA-only tree of *Upham et al., 2019*, one of the most complete and updated phylogenies for mammals. We downloaded the node-dated tree for 4098 mammals, constructed based on a 31-gene supermatrix, from the VertLife website (http://vertlife.org/data/mammals/). We used a pruned version of the tree with the 116 mammalian hosts of Coronaviridae in all analyses in this paper. We kept for each node the 95% credible interval of its age estimate.

## Phylogenetic signal in the association between coronaviruses and mammals

To assess whether closely related coronaviruses interact with similar mammals, and vice-versa, i.e. presence of phylogenetic signal in the association, we used Mantel tests following *Perez-Lamarque et al., 2022b*. Mantel tests were constructed by taking the Pearson correlation between phylogenetic distances and ecological distances. Phylogenetic distances of coronaviruses were computed on the BEAST2 MCC phylogeny. Ecological distances were calculated based on the interaction network matrix containing the association between coronavirus' sOTUs and mammals, accounting for the evolutionary relationships among interaction partners using UniFrac distances (*Chen et al., 2012*). Firstly, we conducted Mantel tests permuting the identity of species but keeping the number of partners per species constant; this allows for assessing the effect of species identity while controlling for the confounding effect of the number of partners. Then, we evaluated the phylogenetic signal in the number of partners alone. Lastly, we calculated clade-specific Mantel tests for sub-networks containing at least 10 species (*Perez-Lamarque et al., 2022b*) to evaluate whether phylogenetic signal was stronger for specific subclades of mammals or coronaviruses. Ten thousand permutations were used in each analysis to assess significance. Analyses were conducted using the phylosignal_network and phylosignal_sub_network functions in the R package RPANDA (*Morlon et al., 2016*).

## Coronaviridae origination and host switches

We used the amalgamated likelihood estimation–(ALE – *Szöllősi et al., 2013*) to reconcile the mammal and coronaviruses evolutionary history using events of cospeciations, host switches, duplications, and losses. Originally designed in the context of gene tree – species tree reconciliations (*Szöllősi et al., 2013*), ALE has also been particularly useful in the context of host-symbiont cophylogenetic analyses as it considers both phylogenetic uncertainty of the symbiont evolutionary history and undersampling of host species (*Perez-Lamarque and Morlon, 2022*; *Groussin et al., 2017*; *Bailly-Bechet et al., 2017*). ALE indeed assumes that host switches may imply an unsampled or extinct intermediate host lineage (*Szöllosi et al., 2013*). ALE therefore intrinsically accounts for the incompleteness of our dataset, *i.e.* the fact that we only observe a subsample of mammalian species for which coronaviruses have been detected among all the infected species.

We ran ALE with the posterior distribution of phylogenetic trees of coronaviruses generated with PhyloBayes to estimate the maximum likelihood rates of host switches, duplications, and losses of the coronaviruses. We first tried running the 'dated' version of ALE, which accounts for the order of branching events in the host phylogeny, therefore only allowing for time-consistent host switches (i.e. host switches that happen between two contemporary host lineages). However, this led to unrealistic parameter estimates (such as very high loss rates) and ALE was not able to output possible reconciliations, suggesting that the mammalian and Coronaviridae trees are too incongruent to be reconciled with only time-consistent host switches, that is the scenario of codiversification is likely unrealistic. We therefore used the 'undated' version of ALE that only exploits the topology of both the host and the symbiont tree and thus does not constrain the host switches to be time-consistent. ALE generated a total of 5000 reconciliations, from which we extracted the mean number of cospeciations, host switches, duplications, and losses. We also reported the likely origination of coronaviruses in mammals (i.e. the branch in the mammal phylogeny that was first infected by coronaviruses) by computing, for each branch of the mammalian tree, the frequency of reconciliations (among the 5000) that supported an origination in that branch. If a reconciliation requires more cospeciation events and fewer host switch events, than expected under a null scenario of independent evolution, this indicates that the evolution of the symbiont was not independent of that of the host, and in this case, we talk about a 'significant reconciliation' (*Dorrell et al., 2021*).

We evaluated the significance of the reconciliation by comparing the estimated number of cospeciation and host switch events to null expectations obtained with ALE by shuffling the mammal host species across the mammal tree, both randomly or within major biogeographic regions according to the proposal of regions by *Barry Cox, 2001* for mammals (six biogeographic regions: North American, South American, African, Eurasian, Oriental, and Australian). We considered a reconciliation to be significant if the observed number of cospeciations was higher than 95% of the null expectations and if the number of host switches was lower than 95% of the null expectations (*Perez-Lamarque and Morlon, 2022*). The likeliness of a host switch between two mammal lineages is measured as the

frequency of the reconciliations in which it occurs. Finally, we reported the ratio of time-inconsistent host switches by focusing on 'back-in-time' switches, from a donor mammal lineage to an older receiver mammal lineage that never coexisted. We identified time-inconsistent host switches directly on the consensus mammalian phylogeny, or considering the 95% credible interval around each age node estimate to avoid counting time-inconsistent host switches that may arise from incorrect estimates of divergence times.

Because ALE estimated a large proportion of time-inconsistent host switches (see Results), we first tested the scenario of a more recent origination by collapsing all mammalian nodes anterior to X Myr into a polytomy at the root of the phylogeny (with X varying from 55 Myr to 5 Myr), such that the coronavirus origination and host switches inferred by ALE could not involve mammal lineages older than X Myr. Second, we investigated the scenario of diversification by pure preferential host switches of the coronaviruses among extant mammals. To do so, we ran ALE on a star mammalian phylogenetic tree. In this context, ALE could no longer infer cospeciations, and only fit events of host switches, duplications, or losses. When inferring a likely host switch between two specific mammalian lineages on a star phylogeny, there are often as many reconciliations suggesting one directionality of the host switch (i.e. from one of the lineages to the other) as the other. We then only kept host switches present in at least 10% of the reconciliations and looked at the ratio between the number of host switches that were estimated within versus between mammal orders. We compared this ratio to a null expectation obtained by randomly shuffling the host mammal species.

We conducted the same set of analyses on the sub-datasets formed by alpha- and betacoronaviruses separately. The dated version of ALE was able to output reconciliations for the alphacoronaviruses dataset, but not for the betacoronaviruses.

Recombination is frequent in viruses and the palmprint region may be recombined, such that different fragments of the palmprint region may have different evolutionary histories, potentially biasing our inference. We carried several recombination tests (OpenRDP - *Martin et al., 2021*), our own custom code, and Gubbins, 78 that were inconclusive, suggesting that the palmprint region is too short to infer anything about recombination. We therefore instead tested whether the results we obtained on the whole 150-amino acid palmprint region could be impacted by recombination by replicating the ALE analyses on two sub-regions: the first part (positions 1–75) and the last part (positions 76–150).

We also repeated our cophylogenetic analyses using eMPRess (*Santichaivekin et al., 2021*), another event-based cophylogenetic approach that reconciles host-symbiont evolutionary histories using maximum parsimony. eMPRess is a recent improved version of the popular Jane approach *Conow et al., 2010*; it differs from Jane especially by not relying only on a heuristic and therefore guarantying that the solution truly corresponds to the maximum parsimony reconciliation(s) (*Santichaivekin et al., 2021*). However, contrary to Jane, eMPRess does not offer the possibility to constrain host switches to occur only among lineages from pre-specified time periods. eMPRess requires specifying cost values for the events of host switches (t), duplications (d), and losses (l). We tested two sets of cost values: (*Alekseev et al., 2008*) cost values that disadvantage host switches (d=6, t=6, l=1) and (*De Groot, 2022*) uniform cost values that favor host switches (d=1, t=1, l=1). As with ALE, we evaluated the significance of the reconciliations using permutations. We ran eMPRess analyses on a set of 50 trees randomly sampled from the posterior distribution of PhyloBayes.

## Sampling biases

More effort has been put on the characterization of coronaviruses associated with humans, domesticated animals, and bats (bats represent 47% of the mammalian species in our dataset, in part because many bat species have been intensively screened for viruses). This can lead to two main sampling biases that could potentially impact our conclusions: (i) unequal sampling effort across mammalian species that have been screened, and (ii) unequal screening of mammalian species across the mammal tree of life. To assert whether such sampling biases could generate spurious results we designed two subsampling strategies and re-run the PhyloBayes and ALE analyses on each subsampled dataset. First, For the unequal sampling effort across mammalian species, we randomly subsampled only three Genbank accession codes per host species (and removed species described by less than three accessions). We replicated the subsampling 50 times and ran ALE on each subsampled dataset. The resulting dataset contained a total of 46 host species. Second, for the unequal screening of mammalian

species across the mammal tree of life, we subsampled the dataset at the level of mammalian orders. Thus, we randomly subsampled up to 10 species per mammalian order. Only four mammalian orders (Chiroptera, Rodentia, Artiodactyla, and Carnivora) had at least 10 species, the other orders were kept unmodified. We replicated the subsampling 50 times and ran ALE on each subsampled dataset. The resulting dataset contained a total of 53 mammalian species.

## Simulation analyses

By running the undated version of ALE either on the mammal phylogeny or a star phylogeny, we proposed a framework to evaluate whether the cophylogenetic pattern is due to a history of ancient codiversification (i.e. a mix of cospeciations, host switches, duplications, and losses; *Figure 1A*) or to a scenario where the coronaviruses diversify more recently by preferential host switches (*de Vienne et al., 2013*; *Figure 1B*). To validate the interpretation of our ALE results, we performed simulations under the two alternative scenarios of codiversification and diversification by preferential host switches.

For the scenario of codiversification, we assumed that coronaviruses originated in the ancestors of bats and that they subsequently codiversified with the mammals by experiencing events of cospeciations, host switches, duplications, and losses. We used the function *sim_microbiota* in the R-package HOME to obtain the corresponding coronavirus sequences and coronavirus-mammals associations (*Perez-Lamarque and Morlon, 2019*). We sampled a number of host switches uniformly between 100 and 150, used a duplication rate of 0.001, and simulated losses by host switching replacement. Given the simulated coronaviruses phylogenies, we simulated on it the evolution of a 450 bp nucleotide sequence using a K80 model with a substitution rate of 0.5 and an expected proportion of variable sites of 50% (see *Perez-Lamarque and Morlon, 2022*, for further details on the model).

For the scenario of coronaviruses diversification by preferential host switches, we used a birth-death model (pbtree function in the R-package phytools) to simulate a phylogenetic tree of the coronaviruses: in our model, each coronavirus lineage is associated with a single host species, a birth event corresponds to a host switch (at rate 50), while a death event corresponds to a loss of a coronavirus in a host lineage (at rate 5). We started the diversification by assuming a single coronavirus infection in *Eidolon helvum* (a bat host of external lineages within betacoroviruses, u25738 and u27845). Then, following *de Vienne et al., 2007* and *Perez-Lamarque et al., 2022a*, we modeled preferential host switches by assuming that for a host switch from a given donor mammal species, each potential receiver species has a probability proportional to exp(–0.035*$d$) where $d$ is the phylogenetic distance between the donor and receiver species. Finally, we simulated RNA sequences of the coronavirus sequences using the function *simulate_alignment* in HOME. For each type of simulation, we generated 50 simulated datasets of mammal-coronavirus associations. For each dataset, we ran PhyloBayes and ALE on both the mammalian phylogeny and the star phylogeny.

In addition, we used simulations to test for a scenario where coronaviruses originated outside of bats but diversify faster within-bats than within other mammalian lineages, as previously suggested given the efficient immune systems of bats (*Banerjee et al., 2020*). We simulated originations in rodents followed by diversification by preferential host switches (as above) with a host switch rate twice more important in bats (rate 80) than in other mammals (rate 40). For each simulated dataset, we ran PhyloBayes and ALE on the star phylogeny and reported the percentage of originations incorrectly inferred in bats. Parameters of the simulations were chosen to mimic the diversity of the original mammal-coronavirus associations.

## Geographic distribution of coronaviridae

We downloaded geographic range maps for each mammalian host species, with the exception of *Homo sapiens*, from the Map of Life website (https://mol.org/species/; see *Marsh et al., 2022*). These maps follow the taxonomy of the Mammal Diversity Database (*Burgin et al., 2018*) supplemented with the Handbook of the Mammals of the World (HMW) database and the Alien Checklist database for invasive species (*Marsh et al., 2022*).

We created a world map with hexagonal, equal-area grid cells of 220 km on which we mapped host and coronavirus species diversity, using the Mollweide world projection to accurately represent areas. At large spatial scales, cells with ~220 km resolution return more reliable diversity estimates than smaller cells (*Hurlbert and Jetz, 2007*). We considered that a host species was present in any given

cell if its range covered at least 30% of the cell area to avoid overestimating diversity. We calculated host species diversity as a simple sum of the species occurring in any given cell, and host phylogenetic diversity as Faith's phylogenetic diversity inde– PD – *Faith, 1992* for each cell. We mapped Coronaviridae diversity using the host-filling method *Pappalardo et al., 2020*: we constructed a range map for each Coronaviridae sOTU by overlapping the range maps of all its hosts. We consider the host filling method appropriate in this case because coronaviruses are obligatory parasites that can only live inside hosts. Next, we calculated Coronaviridae sOTU diversity by summing range maps overlapping on each cell, and Coronaviridae phylogenetic diversity as Faith's PD (*Faith, 1992*). We created these maps in *R Development Core Team, 2018* using the packages epm (*Title et al., 2022*), sf (*Pebesma, 2018*), and ape (*Paradis and Schliep, 2019*).

## Acknowledgements

This work was performed using HPC resources from GENCI-IDRIS (Grants 2021-A0100312405 and 2022-AD010313735). The authors thank B Boussau and the Morlon lab for helpful discussions. RM thanks Campus France and the One Health/Make Our Planet Again Program for a fellowship to conduct this research.

## Additional information

### Funding

| Funder | Grant reference number | Author |
| --- | --- | --- |
| GENCI-IDRIS | 2021-A0100312405 | Benoît Perez-Lamarque Hélène Morlon |
| GENCI-IDRIS | 2022-AD010313735 | Benoît Perez-Lamarque Hélène Morlon |
| One Health/Make Our Planet Great Again Program | 3--5017338047 | Renan Maestri |

The funders had no role in study design, data collection and interpretation, or the decision to submit the work for publication.

### Author contributions

Renan Maestri, Conceptualization, Data curation, Formal analysis, Funding acquisition, Writing – original draft, Writing – review and editing; Benoît Perez-Lamarque, Anna Zhukova, Conceptualization, Formal analysis, Methodology, Writing – original draft, Writing – review and editing; Hélène Morlon, Conceptualization, Supervision, Funding acquisition, Methodology, Writing – original draft, Writing – review and editing

### Author ORCIDs

Renan Maestri ⬚ https://orcid.org/0000-0001-9134-2943
Benoît Perez-Lamarque ⬚ https://orcid.org/0000-0001-7112-7197

Reviewer #1 (Public review): https://doi.org/10.7554/eLife.91745.3.sa1
Reviewer #2 (Public review): https://doi.org/10.7554/eLife.91745.3.sa2
Author response https://doi.org/10.7554/eLife.91745.3.sa3

## Additional files

### Supplementary files

• Supplementary file 1. Supporting information. (**a**) Box with definitions. Our terminology is as in *Perez-Lamarque and Morlon, 2024*. (**b**) Summary of the results obtained for the ALE reconciliations performed on "sliced mammalian phylogenetic trees", i.e. trees where we only considered the X last

Myr and merged nodes older than X Myr into polytomies in order to (i) test a scenario of more or less recent origination of coronaviruses and (ii) avoid back-in-time transfers toward nodes older than the origination time. All reconciliations are significant (when compared to randomizations shuffling host species labels). Reconciliations with an estimated number of cospeciation events larger than the estimated number of transfer events are in bold. (**c**) Host switches between bats and other mammal orders are less likely than expected by chance: For each type of host switches within or between orders, we reported the inferred number of host switches using ALE (on the star phylogeny) and the expected number of host switches if host switches are equally likely between species (obtained by randomly shuffling the host species names). Because we don't have within-OTUs variations with the palmprint region, the directionality all the recent host switches is not identifiable (resulting in equal proportion in both directions at the mammalian order level). Host switches involving bats (Chiroptera) and other mammal orders are indicated in bold. (**d**) Frequency of host switches inferred from bats to other mammal species, including humans. For each mammal species, we computed the average number per reconciliation of host switches from bats to this mammal species. We only reported here the species presenting >10% of chance to experience at least one host switch from bats. (**e**) Frequency of host switches inferred from any mammal species towards humans. We computed the average number per reconciliation of host switches from each mammal species to humans. We only reported here the species presenting >10% of chance to experience at least one host switch. Host switches from bats are highlighted in bold. (**f**) Summary of the different strategies used to evaluate the robustness of our findings. (**g**) Results are qualitatively similar when running ALE on sub-parts on the palmprint region. We reported here results obtained when running ALE on the star phylogeny on (i) the whole palmprint region (positions 1-150), (ii) the first part of the palmprint region (positions 1-75) or (iii) the last part of the palmprint region (positions 76-150). (**h**) Mammal silhouettes taken from open-to-use sources in https://www.phylopic.org/, detailed credits for authors.

- MDAR checklist
- Source data 1. Dataset containing the association matrix between coronaviruses and their mammalian hosts.

## Data availability

All data analyzed during this study are included in the supporting files. Scripts for running the analyses and for replicating the main results are available on the Open Science Framework: https://osf.io/fhwzr/ (*Perez-Lamarque and Maestri, 2024*).

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
