## [Editor Report · eLife assessment]

Maestri et al report the absence of phylogenetic evidence supporting codiversification of mammalian coronaviruses and their hosts, leading to the **important** conclusion that the evolutionary history of the virus and its hosts are decoupled through frequent host switches. The evidence for frequent host switching, derived from state-of-the-art probabilistic modeling of co-evolution, is **convincing**. The study adds a new perspective to the ongoing debate over the timescale of coronavirus evolution.

---

## [Referee Report · Reviewer #1 (Public review)]

Summary:

In this study, Maestri et al. use an integrative framework to study the evolutionary history of coronaviruses. They find that coronaviruses arose recently rather than having undergone ancient codivergences with their mammalian hosts. Furthermore, recent host switching has occurred extensively, but typically between closely related species. Humans have acted as an intermediate host, especially between bats and other mammal species.

Strengths:

The study draws on a range of data sources to reconstruct the history of virus-host codivergence and host switching. The analyses include various tests of robustness and evaluations through simulation.

Weaknesses:

The analyses are limited to a single genetic marker (RdRp) from coronaviruses, but using other sections of the genome might lead to different conclusions. The genetic marker also lacks resolution for recent divergences, which precludes the detailed examination of recent host switches. Careful and detailed reconstruction of the timescale would be helpful for clarifying the evolutionary history of coronaviruses alongside their hosts.

---

## [Referee Report · Reviewer #2 (Public review)]

Summary:

In their study titled "Recent evolutionary origin and localized diversity hotspots of mammalian coronaviruses," authors Benoît Perez-Lamarque, Renan Maestri, Anna Zhukova, and Hélène Morlon investigate the complex evolutionary history of coronaviruses, particularly those affecting mammals, including humans. The study focuses on unraveling the evolutionary trajectory of these viruses, which have shown a high propensity for causing pandemics, as evidenced by the SARS-CoV2 outbreak.

The research addresses a significant gap in our understanding of the evolutionary dynamics of coronaviruses, particularly their history, patterns of host-to-host transmission, and geographical spread. These aspects are important for predicting and managing future pandemic scenarios.

Historically, studies have employed cophylogenetic tests to explore virus-host relationships within the Coronaviridae family, often suggesting a long history of virus-host codiversification spanning millions of years. However, the team led by Perez-Lamarque proposes a novel phylogenetic framework that contrasts this traditional view. Their approach, which involves adapting gene tree-species tree reconciliation, is designed to robustly test the validity of two competing scenarios: an ancient origination and codiversification versus a more recent emergence and diversification through host switching.

Upon applying this innovative framework to the study of coronaviruses and their mammalian hosts, the authors' findings challenge the prevailing notion of a deep evolutionary history. Instead, their results strongly support a scenario where coronaviruses have a more recent origin, likely in bat populations, followed by diversification predominantly through host-switching events. This diversification, interestingly, seems to occur preferentially within mammalian orders.

A critical aspect of their findings is the identification of hotspots of coronavirus diversity, particularly in East Asia and Europe. These regions align with the proposed scenario of a relatively recent origin and subsequent localized host-switching events. The study also highlights the rarity of spillovers from bats to other species, yet underscores the relatively higher likelihood of such spillovers occurring towards humans, suggesting a significant role for humans as an intermediate host in the evolutionary journey of these viruses.

The research also points out the high rates of host-switching within mammalian orders, including between humans, domesticated animals, and non-flying wild mammals.

In conclusion, the study by Perez-Lamarque and colleagues presents an important quantitative advance in our understanding of the evolutionary history of mammalian coronaviruses. It suggests that the long-held belief in extensive virus-host codiversification may have been substantially overestimated, paving the way for a reevaluation of how we understand, predict, and potentially control the spread of these viruses.

Strengths:

The study is conceptually robust, and its conclusions are convincing.

Weaknesses:

The authors could only use the "undated" model in ALE, with the dated method (which only allows time-consistent transfers) failing on their dataset. The authors did attempt to address this issue in the revision, albeit with limited success.

---

## [Author Response]

The following is the authors’ response to the original reviews.

**Public Reviews:**
**Reviewer #1 (Public Review)**:Summary:In this study, Maestri et al. use an integrative framework to study the evolutionary history of coronaviruses. They find that coronaviruses arose recently rather than having undergone ancient codivergences with their mammalian hosts. Furthermore, recent host switching has occurred extensively, but typically between closely related species. Humans have acted as an intermediate host, especially between bats and other mammal species.Strengths:The study draws on a range of data sources to reconstruct the history of virus-host codivergence and host switching. The analyses include various tests of robustness and evaluations through simulation.Weaknesses:The analyses are limited to a single genetic marker (RdRp) from coronaviruses, but using other sections of the genome might lead to different conclusions. The genetic marker also lacks resolution for recent divergences, which precludes the detailed examination of recent host switches. Careful and detailed reconstruction of the timescale would be helpful for clarifying the evolutionary history of coronaviruses alongside their hosts.

The use of a single short genetic marker (the RdRp palmprint region) from coronaviruses is indeed a limitation. However, this marker is the one that is currently used for routinely delimiting operational taxonomic units in RNA viruses and reconstructing their evolutionary history (Edgar et al. 2022, see also the Serratus project; https://serratus.io/); therefore, we took the conscious decision early on to rely on this expertise. Unfortunately, this marker cannot provide robust timescale reconstructions for coronavirus evolution (previous estimates of coronavirus origin range from around 10 thousand years ago to 293 million years ago depending on modeling assumptions). Only future genomic work across Coronaviridae that will characterize multiple genetic regions with different evolutionary rates will allow us to precisely elucidate the timescale of the evolutionary history of coronaviruses alongside their hosts. In the meantime, we show here that, while the RdRp palmprint region cannot by itself resolve the precise timescale of coronavirus evolution, it strongly suggests, when used along with cophylogenetic approaches, a recent evolutionary origin in bats.

We now further discuss these issues and the perspectives offered by future genomic work on lines 462-485.

**Reviewer #2 (Public Review):**
Summary:In their study titled "Recent evolutionary origin and localized diversity hotspots of mammalian coronaviruses," authors Benoît Perez-Lamarque, Renan Maestri, Anna Zhukova, and Hélène Morlon investigate the complex evolutionary history of coronaviruses, particularly those affecting mammals, including humans. The study focuses on unraveling the evolutionary trajectory of these viruses, which have shown a high propensity for causing pandemics, as evidenced by the SARS-CoV2 outbreak.The research addresses a significant gap in our understanding of the evolutionary dynamics of coronaviruses, particularly their history, patterns of host-to-host transmission, and geographical spread. These aspects are important for predicting and managing future pandemic scenarios.Historically, studies have employed cophylogenetic tests to explore virus-host relationships within the Coronaviridae family, often suggesting a long history of virus-host codiversification spanning millions of years. However, the team led by Perez-Lamarque proposes a novel phylogenetic framework that contrasts this traditional view. Their approach, which involves adapting gene tree-species tree reconciliation, is designed to robustly test the validity of two competing scenarios: an ancient origination and codiversification versus a more recent emergence and diversification through host switching.Upon applying this innovative framework to the study of coronaviruses and their mammalian hosts, the authors' findings challenge the prevailing notion of a deep evolutionary history. Instead, their results strongly support a scenario where coronaviruses have a more recent origin, likely in bat populations, followed by diversification predominantly through hostswitching events. This diversification, interestingly, seems to occur preferentially within mammalian orders.A critical aspect of their findings is the identification of hotspots of coronavirus diversity, particularly in East Asia and Europe. These regions align with the proposed scenario of a relatively recent origin and subsequent localized host-switching events. The study also highlights the rarity of spillovers from bats to other species, yet underscores the relatively higher likelihood of such spillovers occurring towards humans, suggesting a significant role for humans as an intermediate host in the evolutionary journey of these viruses.The research also points out the high rates of host-switching within mammalian orders, including between humans, domesticated animals, and non-flying wild mammals.In conclusion, the study by Perez-Lamarque and colleagues presents an important quantitative advance in our understanding of the evolutionary history of mammalian coronaviruses. It suggests that the long-held belief in extensive virus-host codiversification may have been substantially overestimated, paving the way for a reevaluation of how we understand, predict, and potentially control the spread of these viruses.Strengths:The study is conceptually robust, and its conclusions are convincing.Weaknesses:Despite the availability of a dated host tree the authors were only able to use the "undated" model in ALE, with the dated method (which only allows time-consistent transfers) failing on their dataset (possibly due to dataset size?). Further exploration of the question would be potentially valuable.

Our intuition is that ALE in its “dated” version does not necessarily fail on our dataset due to its size: ALE runs, but it provides unrealistic parameter estimates and is not able to output possible reconciliations, as mentioned in our Material and Methods section. We think this issue is mostly due to the fact that there is no pattern of codiversification: the coronavirus and mammal trees are so distinct that finding a reconciliation scenario between these trees with time-consistent switches is very difficult and ALE fails at estimating an amalgamated likelihood for such an unlikely scenario. We now ran the dated version of ALE independently on the smaller alpha and betacoronaviruses datasets. It still fails on the betacoronaviruses dataset. On the alphacoronaviruses dataset, it does output significant reconciliations, however these reconciliations have a majority of events of transfers and losses, confirming that codiversification is unlikely in this clade.

**Reviewer #3 (Public Review):**
Summary:This work uses tools and concepts from co-phylogenetic analyses to reconstruct the evolutionary and diversification history of coronaviruses in mammals. It concludes that crossspecies transmissions from bats to humans are a relatively common event (compared to bats to other species). Across all mammals, the diversification history of coronaviruses suggests that there is potential for further evolutionary diversification.Strengths:The article uses an interesting approach based on jointly looking at the extant network of coronaviruses-mammals interactions, and the phylogenetic history of both these organisms. The authors do an impressive job of explaining the challenges of reconstructing evolutionary dynamics for RNA viruses, and this helps readers appraise the relevance of their approach.Weaknesses:I remain unconvinced by the argument that sampling does not introduce substantial biases in the analyses. As the authors highlight, incomplete knowledge of the extant interactions would lead to a biased reconstruction of the diversification history. In a recent paper (Poisot et al. 2023, Patterns), we look at sampling biases in the virome of mammals and suggest that is a fairly prominent issue, that is furthermore structured by taxonomy, space, and phylogenetic position. Case in point, even for betacoronaviruses, there have been many newly confirmed hosts in recent years. For organisms that have received less intense scrutiny, I think a thorough discussion of potential gaps in data would be required (see for example Cohen et al. 2022, Nat. Comms).I was also surprised to see little discussion of the differences between alpha and beta coronaviruses - there is evidence that they may differ in their cross-species transmission (see Caraballo et al. 2022 Micr. Spectr.), which could call into question the relevance of treating all coronaviruses as a single, homogeneous group.Some of the discussions in this paper also echo previous work by e.g. Geoghegan et al. (see 2017, PLOS Pathogens), which I was surprised to not see discussed, as it is a much earlier investigation of the relative frequencies of co-divergence and host switches for different viral families, with a deep discussion of how this may structure future evolutionary dynamics.

We totally agree that sampling biases in the virome of mammals is a prominent issue, which is why we conducted a series of sensitivity analyses to test their effect on our main conclusions. We thoroughly tested the effect of (i) the unequal sampling effort across mammalian species that have been screened and (ii) the unequal screening of mammalian species across the mammalian tree of life by subsampling the data to correct for the unequal sampling effort (see Supporting Information Text). In both cases, we still reported low support for a scenario of codiversification, the origin in bats in East Asia, the preferential host switches within mammalian orders, and the rare spillovers from bats to humans. The robustness of our findings to sampling biases may be explained by the fact that the cophylogenetic approach we used (ALE) explicitly accounts for undersampling by assuming that all host switches involve unsampled intermediate hosts. To address the reviewer's comment, we now better underline the importance of sampling biases in our main text (see Discussion, lines 487-494) with supporting references (note that we did not find the Cohen et al. Nature Comm reference). We also better highlight our sensitivity analyses by moving them from the Supporting Information Text to the main text.

We agree that distinguishing between alpha and beta coronaviruses provides useful additional insights. We have run separate cophylogenetic analyses for these two sub-clades and now report the results of these additional analyses in the revised manuscript, and put them in context with the existing literature about the two sub-clades.

We were not aware of the work of Geoghegan et al. (see 2017, PLOS Pathogens), thank you for providing this reference that is now cited.

**Reviewer #1 (Recommendations For The Authors):**
(1) Overall I found this paper to be quite difficult to follow. The text needs clearer structure, which can be helped by writing in shorter paragraphs and adding section headings. For example, there are some very long paragraphs starting on L83, L176, L215, L511, and L598.

We have now added section headings and divided these paragraphs into smaller ones.

(2) It would be helpful to define some of the key terminology relating to the evolutionary interactions between the viruses and their hosts. Some of the terms that are typically used in the context include "coevolution", "cospeciation", "codivergence", and "codiversification". These have different meanings and need to be used carefully. The paper mostly deals with "codivergence" between coronaviruses and their host species.

We now provide a list of definitions in Box S1. These definitions are as in our recent article clarifying the differences between these patterns/processes (Perez-Lamarque & Morlon 2024).

Specific commentsL83-L105: This paragraph can be written more concisely.

We prefer to keep this paragraph like this as it contains key explanations that are necessary for understanding our approach and results.

Figure 1: The timescales of the trees are rather confusing. The different scales are indicated by the gray shading but this is easy to overlook. Maybe stretching or compressing the trees horizontally would help to emphasise the different timescales.

Done.

Figure 2: Note that the maximum clade credibility tree is a specific tree sampled from the posterior distribution - it is not a consensus tree. In the figure caption, the meaning of "location" is unclear.

We have removed the word “consensus”, thank you for noting this. We have replaced “location” by “branching order”.

L461: How was the model chosen, and why were different models used in the BEAST and PhyloBayes analyses?

We did our PhyloBayes analyses first and used the LG model following methodology outlined in previous studies using ALE (e.g. Groussin et al. 2017; Dorrell et al. 2021). Unfortunately, the LG model is not available in the default version of BEAST2 so we had to use a different model (the WAG model). We have now run BEAST2 with the LG model (thanks to the BEAST_CLASSIC package) and we obtained very similar results (see Figure below showing the BEAST consensus trees obtained with the WAG or LG models – they only slightly differ by the branching of the u7351 OTU). We have now added this information in the Methods section.

L477: It is not clear to me how the PhyloBayes and BEAST analyses differ. Please expand the explanation of why PhyloBayes was used here.

We have now clarified this (lines 594-597).

L568: Why not test explicitly for recombination?

We did test for the occurrence of recombination using several approaches, including

OpenRDP (https://github.com/PoonLab/OpenRDP), our own custom code, and Gubbins (Croucher et al. 2015). These tests were however inconclusive, indicating either the absence or presence of recombination, thus suggesting that the palmprint region is too short to infer anything about recombination. We thus do not exclude the possibility that recombination occurred, and test the robustness of our results to recombination by running our analyses on different sub-parts of the palmprint region. We have clarified this in our Material & Methods.

L618: "DNA sequences" -> "RNA sequences"

Done.

The paper contains numerous minor grammatical errors and would benefit from careful proofreading and editing. Please check the use of plurals and apostrophes. Some of the errors are listed below:L49: "As several" -> "As with several"

Done.

L178: "reconciliates" -> "reconciles"?

Done.

L199: "extent" -> "extant"

Done.

L289: This sentence needs rephrasing to avoid a triple negative ("cannot ... reject ... not present")

Done.

L469: "temporary" -> "temporal"

Done.

L470: "neglectable" -> "negligible"

Done.

L577: "not only relying" -> "not relying only"

Done.

**Reviewer #2 (Recommendations For The Authors):**
The study is generally well-constructed and its results are convincing. However, considering the availability of a dated host tree, conducting a dated reconciliation analysis could be beneficial. Creating a smaller sub-dataset and performing a dated reconciliation analysis would likely be a valuable addition to the research.

We have now run the dated version of ALE on both the alpha and betacoronaviruses subclades. ALE dated still does not output reconciliations on the betacoronaviruses dataset, but it does on the smaller alphacoronaviruses dataset. We found significant reconciliations, indicating that mammal-alphacoronavirus associations are not random with respect to phylogeny, but the reconciliations involved more host switch and loss events (38 switches + 29 losses) than cospeciation events (65), indicating cophylogenetic signal in the absence of phylogenetic congruence (Perez-Lamarque & Morlon 2024). We now present the results on lines 264-282.

**Reviewer #3 (Recommendations For The Authors):**
I think the results are written in a very speculative way, with many sentence fragments that should really be part of the discussion.

We have carefully checked our Results section and rephrased or removed formulation that may have been perceived as speculative.

There are a lot of considerations in this manuscript about spread and future pandemics, but I think this is very far from the topic of this paper. When we quantified the coevolutionary risk of bats-betacovs in a recent paper (Forero et al. 2024, Virus Evol.), we only briefly touched upon this discussion because we compared our outputs with a measure of human population density. I don't think the manuscript needs to talk about epidemiology at all, and it would probably be more useful as a purely evo-bio piece.

We think that it is useful to discuss the potential implications of our results for future pandemics, even though we agree that this discussion is rather speculative. We have removed the mention of predictions in the Abstract and have softened our wording in the Discussion.

References:

Croucher, N.J., Page, A.J., Connor, T.R., Delaney, A.J., Keane, J.A., Bentley, S.D., et al. (2015). Rapid phylogenetic analysis of large samples of recombinant bacterial whole genome sequences using Gubbins. Nucleic Acids Res., 43, e15.

Dorrell, R.G., Villain, A., Perez-Lamarque, B., Audren de Kerdrel, G., McCallum, G., Watson, A.K., et al. (2021). Phylogenomic fingerprinting of tempo and functions of horizontal gene transfer within ochrophytes. Proc. Natl. Acad. Sci., 118, e2009974118.

Edgar, R.C. et al. (2022). Petabase-scale sequence alignment catalyses viral discovery. Nature 602, 142–147.

Groussin, M., Mazel, F., Sanders, J.G., Smillie, C.S., Lavergne, S., Thuiller, W., et al. (2017).

Unraveling the processes shaping mammalian gut microbiomes over evolutionary time. Nat. Commun., 8, 14319.

Perez-Lamarque, B. & Morlon, H. (2024). Distinguishing cophylogenetic signal from phylogenetic congruence clarifies the interplay between evolutionary history and species interactions. Syst. Biol.